# Improved Deep Learning Predictions for Chlorophyll Fluorescence Based on Decomposition Algorithms: The Importance of Data Preprocessing



Lan Wang [1,2,3], Mingjiang Xie [2], Min Pan [4,*], Feng He [4], Bing Yang [5], Zhigang Gong [2], Xuke Wu [1], Mingsheng Shang [2] and Kun Shan [2,*]

1   School of Computer Science and Technology, Chongqing University of Posts and Telecommunications, Chongqing 400065, China; wanglan@cque.edu.cn (L.W.); xuke_wu@163.com (X.W.)
2   Chongqing Institute of Green and Intelligent Technology, Chinese Academy of Sciences, Chongqing 400714, China; xiemingjiang@cigit.ac.cn (M.X.); gongzhigang@cigit.ac.cn (Z.G.); msshang@cigit.ac.cn (M.S.)
3   School of Artificial Intelligence, Chongqing University of Education, Chongqing 400065, China
4   Dianchi Lake Ecosystem Observation and Research Station of Yunnan Province, Kunming Dianchi and Plateau Lakes Institute, Kunming 650228, China; hefengyunnan@163.com
5   Chongqing Eco-Environmental Monitoring Center, Chongqing 401147, China; cq_yangbing@163.com
*   Correspondence: panmin333@foxmail.com (M.P.); shankun@cigit.ac.cn (K.S.)

**Abstract:** Harmful algal blooms (HABs) have been deteriorating global water bodies, and the accurate prediction of algal dynamics using the modelling method is a challenging research area. High-frequency monitoring and deep learning technology have opened up new horizons for HAB forecasting. However, the non-stationary and stochastic process behind algal dynamics monitoring largely limits the prediction performance and the early warning of algal booms. Through an analysis of the published literature, we found that decomposition methods are widely used in time-series analysis for hydrological processes. Predictions of ecological indicators have received less attention due to their inherent fluctuations. This study explores and demonstrates the predictive enhancement for chlorophyll fluorescence data based on the coupling of three decomposition algorithms with conventional deep learning models: the convolutional neural network (CNN) and long short-term memory (LSTM). We found that the decomposition algorithms can successfully capture the time-series patterns of chlorophyll fluorescence concentrations. The results indicate that decomposition-based models can enhance the accuracy of single models in predicting chlorophyll concentrations in terms of the improvement percentages in RMSE (with increases ranging from 25.7% to 71.3%), MAE (ranging from 28.3% to 75.7%), and $R^2$ values (increasing ranging from 14.8% to 34.8%). In addition, the comparison experiment for different decomposition methods might suggest the superiority of singular spectral analysis in hourly predictive tasks of chlorophyll fluorescence over the wavelet transform and empirical mode decomposition models. Overall, while decomposition methods come with their respective strengths and weaknesses, they are undeniably efficient in combination with deep learning models in dealing with the high-frequency monitoring of chlorophyll fluorescence data. We also suggest that model developers pay more attention to online data preprocessing and conduct comparative analyses to determine the best model combinations for forecasting algal blooms and water management.

**Keywords:** chlorophyll fluorescence; deep learning; online monitoring; time-series decomposition; Dianchi Lake



## 1. Introduction

Harmful algal blooms (HABs) have become a worldwide severe environmental problem by releasing excess toxins, which can have detrimental effects on aquatic ecosystems and endanger human health [1]. The timely and high-accuracy prediction of HAB occurrence and intensity is essential in controlling their detrimental environmental and public

health effects [2]. Continuous and high-frequency monitoring technologies are widely applied in HAB monitoring. For instance, flow cytometry analysis [3], hyperspectral imagery [4] and unmanned aerial vehicles [5] could monitor the real-time distribution of HABs. In particular, chlorophyll-*a* (Chl*a*) is a crucial parameter for characterizing phytoplankton communities, making it a commonly employed diagnostic pigment in measurements [2]. Chlorophyll fluorescence sensors can provide fast, cost-effective, and highly temporal revolution data to train the advanced models for HAB predictions [6]. However, high-resolution time-series data often exhibit stochastic non-stationarity distributions, owing to environmental drivers interactively influencing the formation of HABs [7]. Therefore, additional modelling efforts are required to overcome the challenges associated with HAB forecasting.

In recent years, data-driven models have gained wider usage for forecasting algal blooms in inland water [8]. Machine learning algorithms, including neural networks [9], evolutionary computation [10], support vector machines (SVMs) [11], random forests [12], and gradient boost machines [13], are well-known to be helpful in predicting HABs. Consequently, the non-linearity and intermittency process behind the algal monitoring data hinders the performance of machine learning in the accurate forecasting of early-stage blooms [14]. With evolving artificial intelligence, deep learning (DL) models have received increasing attention in HAB forecasts [15]. For instance, the independent recurrent neural network (RNN) [16], long short-term memory (LSTM) [17], and gated recurrent unit (GRU) [18] have been increasingly used to forecast HABs [19,20]. Some researchers have also demonstrated the effectiveness of image-based convolutional neural networks (CNNs) in modelling HABs [21,22]. Nonetheless, deep learning requires large-scale monitoring to train models. In fact, the online monitoring of HAB sensors often contains abnormal values, peak values, and error components with irregular random movements, causing inherently non-linear, complex, and non-stationary algal sequence time series [23]. Therefore, pure data-driven methodologies prove unsatisfactory in addressing the high variations in algal dynamics [24]. In this case, choosing an appropriate data preprocessing procedure might be essential to increase the forecasting accuracy of the predictive algal parameters [25,26].

The decomposition-based frameworks have been demonstrated to extract the dynamic features of time-series data and enhance models' predictive performance in an increasing number of studies [27]. In contrast to conventional time-series models, the decomposition-based frameworks divide the time series into components of varying frequencies, predicting each component individually with deep learning models, and summing them to obtain the predicted results. From the perspectives of decomposition algorithms, the wavelet transform (WT) and empirical mode decomposition (EMD) methods are two of the most commonly used decomposition methods [28]. For example, Liu et al. [26] proposed a hybrid prediction model combining WT and LSTM, which decomposes the original algal parameters series into multiple sub-series using wavelet transform, and then employs LSTM on the sub-series components. Zhu et al. [29] reported the hybrid EMD-LSTM with the attention mechanism model and indicated that empirical mode decomposition (EMD) can effectively enhance the smoothness of the time series and increase predictive accuracy. In addition, Luo et al. [30] develop an improved empirical mode decomposition model (EEMD)–LSTM prediction model to predict water quality. Apaydin et al. [31] investigate the singular spectral analysis (SSA) with LSTM to increase the monthly streamflow prediction accuracy. Following that, integrated SSA and genetic-based models have been developed for river flow forecasting and showed improved accuracy [32]. To date, no reported study has compared the difference in employing these above-mentioned decomposition-based methods in dealing with high-frequency water-quality data, especially for chlorophyll fluorescence monitoring.

In this study, we evaluate and compare the performance of different hybrid approaches that couple WT, EEMD, and SSA decomposition for extracting the sub-series component of Chl*a* data, along with the deep learning approaches for predicting HABs. The Chl*a* is obtained from a substantial volume of in situ multi-sensor monitoring data in Lake Dianchi, China. Specifically, the goals of this paper are as follows: (1) to obtain better prediction

performance of the Chl*a* fluorescence by combining the WT, EEMD, and SSA decomposition approaches; (2) to develop decomposition-based hybrid models and evaluate the prediction effectiveness of the models; and (3) to further compare the prediction performance of different decomposition approaches for chlorophyll fluorescence forecasting. This study demonstrates that the decomposition-based Chl*a* prediction methods could function as a robust and trustworthy tool for forecasting HABs in water management.

## 2. Materials and Methods

### 2.1. Study Sites and Data

The study area is located in Lake Danchi (24°40′–25°02′ N, 102°36′–102°47′ E), Kunming, Yunnan, in southwest China (Figure 1). Lake Danchi is a famous plateau freshwater lake with a surface area of 330 km², and the average depth measures 4.4 m, with a maximum water depth reaching 6.7 m and a watershed area spanning 2920 km² [33]. Additionally, an artificial causeway, Haigeng Dam, divides the lake into two parts—Caohai in the north and Waihai in the south—covering 286.78 km² [34]. Over recent decades, this eutrophic lake has frequently experienced intense HABs and has gradually developed to have some of the heaviest cyanobacterial blooms in Chinese lakes [35]. Due to the low water flow and high pollutant concentration, cyanobacterial blooms are a frequent occurrence in Caohai, which is located closer to the urban area. This study collected the average values of the Chl*a* data of the Duanqiao and Caohai center sampling sites through continuous monitoring every four hours from 1 February 2019 to 8 January 2021 (Table 1). The two online monitoring sites in Caohai, Dianchi Lake, belong to the sections managed by Kunming Municipal Environmental Monitoring Center, Yunnan, China. The first 50% of the sequence (February 2019–January 2020) was used to train the hybrid models. The remaining data (January 2020–January 2021) were eventually used to test the model performance.

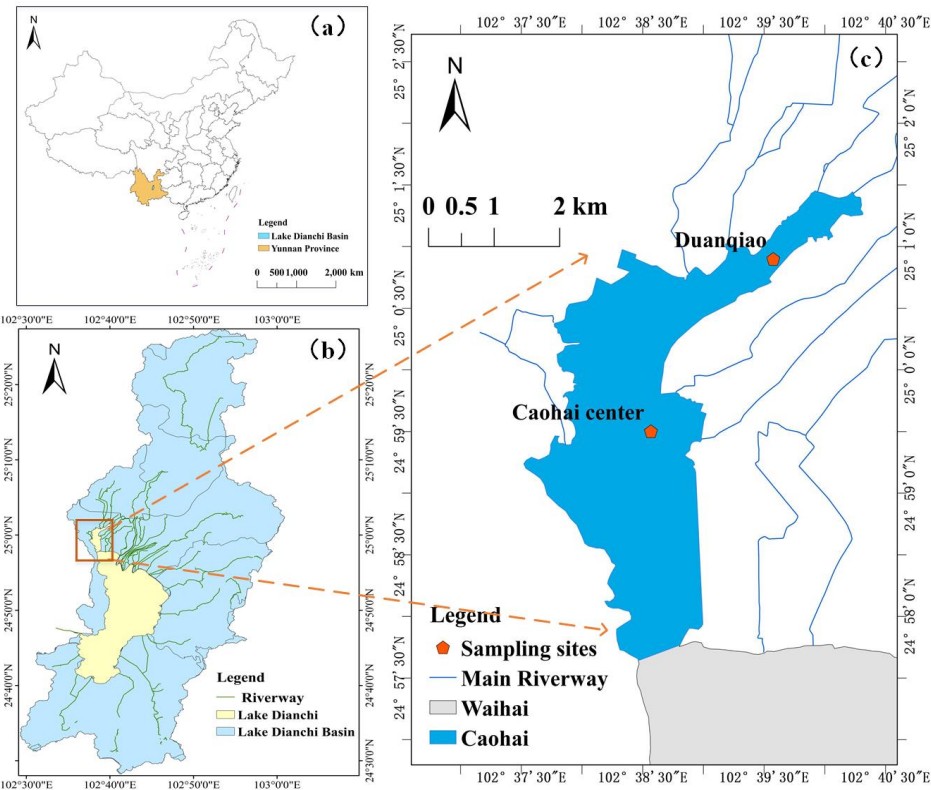

**Figure 1.** Overview of the study area in Caohai, Lake Dianchi, and the distribution of sampling sites. (**a**) Location of Lake Dianchi in Yunnan, Southwest China. (**b**) Distribution of Lake Dianchi and riverway networks in Lake Dianchi Basin. (**c**) The red pentagon marks the location of the two sampling sites inside Caohai of Lake Dianchi.

**Table 1.** Overview of online monitoring datasets for chlorophyll-*a* at two monitoring sites.

| Location | Sample Size | Mean | Standard Deviation | Min | Median | Max |
|---|---|---|---|---|---|---|
| Duanqiao | 4081 | 8.46 | 6.19 | 0.32 | 6.95 | 55 |
| Caohai Center | 4081 | 14.41 | 9.44 | 0.49 | 13 | 60 |

*2.2. Decomposition-Based Deep Learning Model Development*

In this section, we present the multi-decomposition architecture (Section 2.2.1) and introduce the wavelet transformation analysis (Section 2.2.2), ensemble empirical mode decomposition (Section 2.2.3), and singular spectral analysis (Section 2.2.4). We further illustrate convolutional neural networks (Section 2.2.5) and LSTM (Section 2.2.6).

2.2.1. The Multi-Decomposition Architecture

This study established and compared the performance of three decomposition-based models in forecasting the Chl*a* in the Caohai center and Duanqiao sites of Lake Dianchi. A technical flowchart of this study is presented in Figure 2. Firstly, the original series data of the Chl*a* were decomposed into sub-series based on the multi-decomposition process. Then, the sub-sequences were input into deep learning models to be trained and validated one by one. Finally, all the individual forecasted sub-series were summed to derive the predicted results of the sequence of Chl*a*.

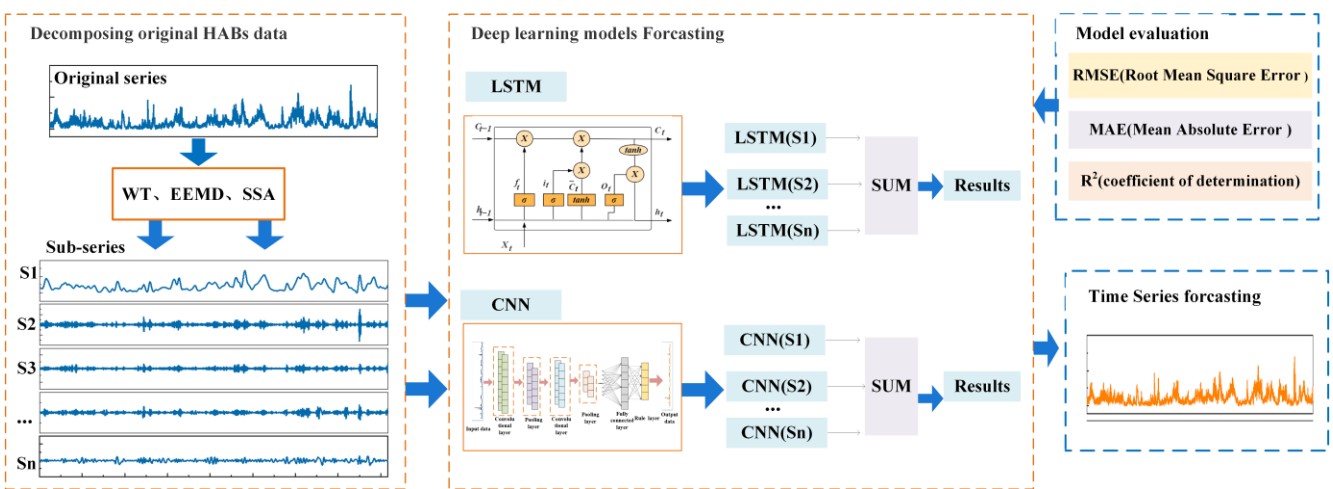

**Figure 2.** Schematic flow chart of the decomposition-based hybrid deep learning models.

2.2.2. Wavelet Transformation Analysis

The wavelet transformation analysis method (WT) is a useful mathematical tool of signal analysis theory in physics and engineering [36,37]. By decomposing the original signal into several sub-components at different time frequency spaces, the WT can analyze non-stationary data and effectively extract time frequency features of the original time series simultaneously. The sub-sequences are typically derived from a template referred to as the "mother wavelet", and these deconstructed wavelets are scaled and translated based on the mother wavelet. The advantage of wavelet analysis lies in its capacity for the adaptable selection of the mother wavelet to match the specific characteristics of the investigated time series. However, determining scale and translation parameters for every possible position necessitates significant computational effort when utilizing a continuous wavelet transformation (CWT).

In contrast, the discrete wavelet transformation (DWT) substantially alleviates the computational complexities associated with wavelet transformations by adopting dyadic

scales and positions, typically based on powers of two [26]. The DWT of a time series, $f(t)$, is typically carried out as follows:

$$W_f(a,b) = \sum_{a,b \in Z} f(t) \Psi^*_{a,b}(t) \tag{1}$$

$$\Psi^*_{a,b}(t) = m_0^{-\frac{a}{2}} \Psi(m_0^{-a}t - n_0 b), m_0 > 1, n_0 > 0 \tag{2}$$

where the integers $a$ and $b$ represent the decomposition level and translation factor, respectively; the constant $m_0$ is decomposition scale factor; the constant $n_0$ is the position factor of translation; $\Psi^*_{a,b}(t)$ is the wavelet function; $\Psi(t)$ is the mother wavelet that can be set as the "Daubechies", "Haar", and "Morlet" wavelets; and $W_f(a,b)$ are the DWT coefficients. The discrete wavelet transformation (DWT) employs high-pass and low-pass filters to decompose the original time series, $f(t)$, into different resolution levels, yielding a low-frequency approximation sub-sequence ($A_n$) and a high-frequency sub-sequence ($D_1$, $D_2$, ..., $D_n$), and finally obtaining detailed coefficients and approximation sub-time series.

2.2.3. Ensemble Empirical Mode Decomposition (EEMD)

Empirical mode decomposition (EMD) is a noise-reduction, signal-adaptive decomposition algorithm for non-linear and non-stationary data [38]. The original time series can be decomposed into finite modal components and intrinsic mode functions (IMFs) that contain only a single instantaneous frequency, and residual (Res) [39]. Nonetheless, the noise of signal may result in mode aliasing within the IMFs, consequently generating inaccurate time frequency distributions and diminishing the interpretability of the IMFs. To mitigate the adverse impact of noise during the decomposition process, Zhaohua and Norden (2009) [40] propose ensemble empirical mode decomposition (EEMD), a data analysis approach that incorporates white noise into the original time series. The detailed process of EEMD is as follows:

Given an original series, denoted as $f(t)$, the detailed process of EEMD is described below:

(1)　Add random white noise to the original time series, $n^i(t) \sim (0, \sigma^2)$, where $\sigma$ is known.

$$f^i(t) = f(t) + n^i(t) \tag{3}$$

where $i$ denotes the count of white noise additions.

(2)　The EMD algorithm is employed to decompose the composited sequences, $f^i(t)$, with noise into $IMFs^i_j(t)$, ($j$ = 1, 2, ..., K), and the residual, $Res^i(t)$.

$$f^i(t) = \sum_{j=1}^{K} IMFs^i_j(t) + Res^i(t) \tag{4}$$

where $IMFs^i_j(t)$ indicates the $j$-th IMF component of derived from the decomposition of the $i$-th mixed original series.

(3)　Repeat the steps described above N times, each time using different Gaussian white noise, and determine the corresponding IMFs.

(4)　Repeat the aforementioned procedure N times, introducing different Gaussian white noise in each iteration, and obtain each corresponding IMF. Compute the average of the sum of the corresponding decomposed IMFs over N iterations to mitigate the impact of the introduced white noise on the original signal. The $j$-th IMF component is as follows:

$$avr(f_j(t)) = \frac{1}{N} \sum_{j=1}^{K} IMFs^i_j(t) \tag{5}$$

(5)　Finally, the original series, $f(t)$, are decomposed by EEMD models, which can be expressed as follows:

$$f(t) = \sum_{j=1}^{K} avr(f_j(t)) + Res^i(t) = \frac{1}{N} \sum_{j=1}^{K} IMFs^i_j(t) + Res^i(t) \tag{6}$$

where $i$ = 1, 2, ..., N.

2.2.4. Singular Spectral Analysis

Singular spectral analysis (SSA) is a non-parametric method for estimating the spectral characteristics of time series data, aimed at discerning distinct patterns of variability [41]. The fundamental framework of SSA encompasses data embedding, singular value decomposition (SVD), eigentriple grouping, and diagonal averaging.

Embedding can be viewed as a transformation that converts a one-dimensional time series into a trajectory matrix using the selected window length (L), upon which SVD is performed. The window length is the time step by which the data is further divided to extract components. The final product of singular value decomposition is feature triples, the count of which matches the chosen window length. Consider $Y_N = (y_1, y_2, \ldots, y_N)$, which is not a series with all zeros; let "X" be the trajectory matrix, which can be expressed as

$$X = \begin{pmatrix} y_1 & y_2 & y_3 & \cdots & y_K \\ y_2 & y_3 & y_4 & \cdots & y_{K+1} \\ y_3 & y_4 & y_5 & \cdots & y_{K+2} \\ \vdots & \vdots & \vdots & \ddots & \vdots \\ y_L & y_{L+1} & y_{L+2} & \cdots & y_N \end{pmatrix} \tag{7}$$

where $K = N - L + 1$. Note that the resulting trajectory matrix is a Hankel matrix, implying that all elements along the diagonal, where i + j is a constant, are equivalent.

Secondly, singular value decomposition is applied to the trajectory matrix, X. Let $S = XX^T$, $\lambda_1, \lambda_2, \ldots, \lambda_L$ is the eigenvalue of $S$ sorted in descending order ($\lambda_1 \gg \ldots \lambda_L \gg 0$), and $U_1, \cdots U_L$ are the standard orthonormal vectors of the matrix $S$ corresponding to these eigenvalues. Let $d = \text{rank}(X) = \max\{i, \lambda_i > 0\}$ (in practical sequences, usually $d = L^*$, $L^* = min(L, K)$). Then, the SVD of the trajectory matrix can be written as $X = X_1 + \ldots + X_d$, where $X_i = \sqrt{\delta_i} U_i V_i^T$.

In the grouping step, we can choose to analyze the periodogram, right eigenvector scatter plot, or eigenvalue function plot to distinguish between noise and signal. In the process of reconstructing the signal, there are no specific rules for the way of grouping. The set of subscripts $\{1, \ldots, d\}$ can be divided into $m$ disjoint subsets according to the properties of the time series to be reconstructed, i.e., $I_1, I_2, \ldots, I_m$. If $I = \{i_1, \ldots, i_p\}$, then the composite matrix corresponds to $X = X_{I1} + X_{I2} + \ldots + X_{IM}$. The final step of SSA involves transforming each resulting matrix from the grouping into a new sequence of length N. Let $T$ be a $L \times K$ matrix with elements $t_{ij}$, $1 \leq i \leq L$, $1 \leq j \leq K$. Set $L^* = min \{L, K\}$, $K^* = max \{L, K\}$, and $N = L + K - 1$. Let $t_{ij}{}^* = t_{ij}$ if $L < K$, and $t_{ij}{}^* = t_{ji}$ otherwise. Through the process of diagonal averaging, matrix $T$ is transformed into a series $t_1, t_2, \ldots, t_N$ using the following formula:

$$t_k = \begin{cases} \frac{1}{k} \sum_{m=1}^{k} t^*_{m,k-m+1}, & 1 \leq k < L^* \\ \frac{1}{L^*} \sum_{m=1}^{L^*} t^*_{m,k-m+1}, & L^* \leq k < K^* \\ \frac{1}{N-k+1} \sum_{m=k-K^*+1}^{N-K^*+1} t^*_{m,k-m+1}, & K^* \leq k < N \end{cases} \tag{8}$$

A single RCt sub-sequence of length N can be obtained according to the formula. The new X component is the sum of d RCt components and can be expressed as $X^* = RC_1 + RC_2 + \ldots + RC_d$.

2.2.5. Convolutional Neural Network

One-dimensional convolution (1D-CNN) can be executed by employing a filter that is specialized for handling sequential data, allowing for the extraction of sequence features as the network slides over the data using convolution kernels. A standard 1D-CNN typically comprises an input layer, multiple interleaved convolutional and pooling layers, a fully

connected layer, and an output layer, as illustrated in Figure 3. The convolutional and pooling layers are distinctive components of the convolutional neural network. With a single-layer 1D-CNN, the output vector Y : $\left[ y_1, \cdots, y_j, \cdots, y_{\frac{m-n+1}{t}} \right]$ is obtained as follows:

$$y_j = \begin{cases} \sum\limits_{i=1}^{b} I_{j+i-1} k_i, & j = 1 \\ \sum\limits_{i=1}^{b} I_{j+i+\tau-2} k_i, & 1 < j \leq \frac{a-b+1}{t} \end{cases} \tag{9}$$

where $I:[I_1, \ldots, I_i, \ldots, I_a]$ is an input vector with the size of *a*, *t* is the stride, and *b* stands for the size of the convolution kernel, *k*. After $y_1$ is obtained according Equation (3), the calculation window slides back to $I_{t+1}$ to calculate. This process is repeated until there are no remaining data from the input.

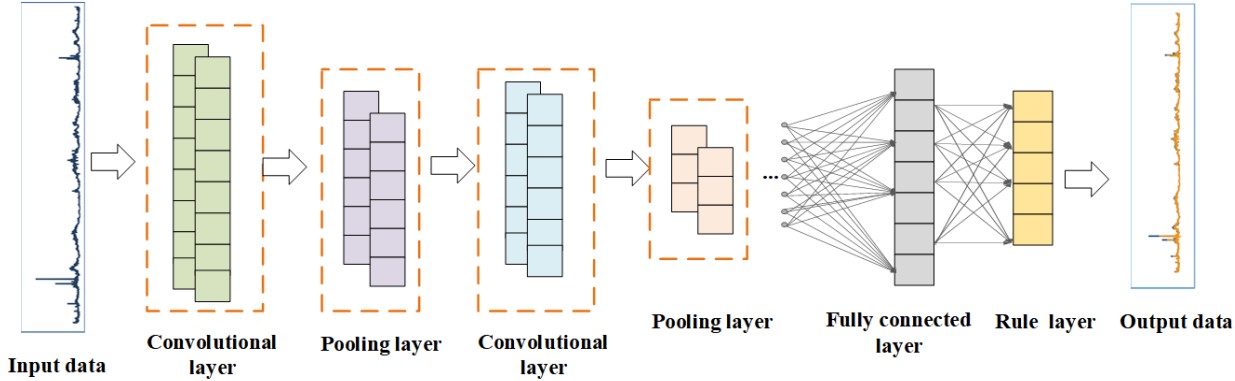

**Figure 3.** One-dimensional CNN model flowchart.

### 2.2.6. Long Short-Term Memory

The LSTM network is variant of a recurrent neural network (RNN) model and is improved on the basis of recurrent neural networks to make it have a long short-term memory function. The LSTM effectively captures long-range dependencies, addressing the problems of gradients exploding and vanishing during backpropagation that are common in traditional RNNs [17]. Each LSTM block consists of a memory cell and three parts: the input gate, forget gate, and output gate (Figure 4). The manner in which information from the previous layer is passed to the current layer is determined by each gate selectively. The memory cell acts as an accumulator of state information, preserving the hidden details of the time series. This allows the LSTM to leverage long-term historical context. The specific content is as follows: In LSTM, the forgetting gate first determines the retention of the state at the previous moment, and the calculation formula is

$$f_t = \sigma \left( W_f \cdot [h_{t-1}, \, x_t] + b_f \right) \tag{10}$$

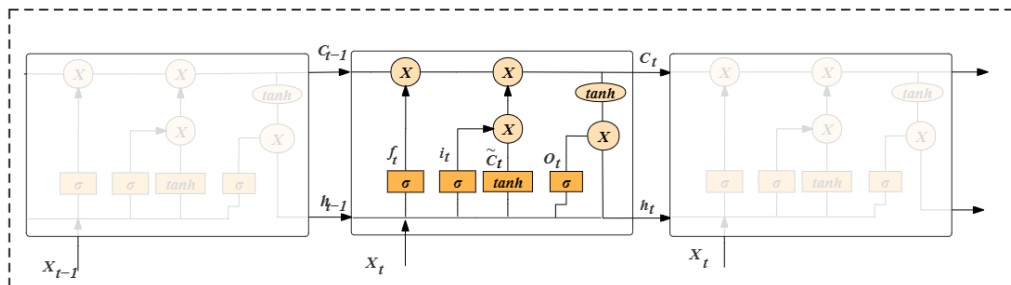

**Figure 4.** LSTM model flowchart.

In this formula, $\sigma$ is the activation function Sigmoid, $W_f$ represents the weights of forgotten gate weights, and $b_f$ represents the bias of the forget gate. The Sigmoid function maps the input and the state of the previous moment to a value from 0 to 1. The value of $f_t$ is 1 to indicate full retention and 0 to signify complete discarding. The input gate determines the extent to which the current network input, denoted as $x_t$, is incorporated into the cell state $C_t$.

$$i_t = \sigma \left( W_i \cdot [h_{t-1},\ x_t] + b_i \right) \tag{11}$$

$$\widetilde{C}_t = tanh(W_c \cdot [h_{t-1},\ x_t] + b_c) \tag{12}$$

$$C_t = f_t \times C_{t-1} + i_t \times \widetilde{C}_t \tag{13}$$

$W_i$ and $b_i$ are the weights and bias of the input gate; $W_c$ and $b_c$ represent the weight and bias when constructing the candidate vector, determining the proportion of forgetting by the sigmoid function. $C_t$ of Equation (13) implements the cell state update at moment $t$. The output gate needs to determine the output value with the following formula:

$$o_t = \sigma \left( W_o \cdot [h_{t-1},\ x_t] + b_o \right) \tag{14}$$

$$h_t = o_t \cdot tanh(C_t) \tag{15}$$

where $W_o$ and $b_o$ are the weights and bias of the output gate. The current state, $C_t$, is multiplied by the output, $o_t$, of the activation function layer after tanh to obtain the output, $h_t$, at the current moment.

*2.3. Model Implementation*

In this study, decomposition-based hybrid models are employed for predicting the concentration of Chl*a* in Dianchi Lake. To appropriately train the deep learning predicting models, we split the modeling procedure into training and testing. In the training part, the first 50% of the decomposed sequence (February 2019–January 2020) are input to build the network. In the testing part, the remaining data (January 2020–January 2021) are accustomed to estimate the model performance. Based on the training data, to prevent the influence of varying scales on parameter learning, Chl*a* concentrations are normalized to a range from 0 to 1 using min–max normalization. Furthermore, we utilize the mean square error (MSE) as the loss function and implement the adaptive momentum estimation method (Adam) to optimize the weights. We implemented our deep learning network on the Keras development platform. In this study, the Daubechies-4 (db4) mother wavelet was utilized to perform a three-level decomposition of the original time series, a choice popular for its widespread acceptance and efficient performance [26]. The ensemble number of the EEMD model was set to 100, and the standard deviation of Gaussian white noise, $n^i(t)$, was 0.05 [30]. We set the window length of the SSA to 15 according to the empirical evaluation of component contributions in the experiment. The WT, EEMD, and SSA were carried out using the MATLAB R2019b software. For the purpose of conducting an equitable comparative analysis across various decomposition methodologies, the parameters for the CNN and LSTM are presented in Table 2.

**Table 2.** The hyper-parameters of the CNN and LSTM models.

| Parameter Name | CNN | LSTM |
|---|---|---|
| Time Lag | 10 | 10 |
| Hidden Size | 16 | 16 |
| Learning Rate | 0.001 | 0.002 |
| Epoch | 100 | 50 |
| Batch Size | 32 | 32 |
| Activation Function | relu | relu |
| Kernel_Size | 3 | / |
| Input_size | 1 | 1 |

*2.4. Evaluation Metrics*

To comprehensively measure the prediction performance of the decomposed-base hybrid deep learning models, three different criteria are used, including the RMSE (root mean square error), mean absolute error (MAE), and the coefficient of determination ($R^2$). The formulas for calculating these indicators are as follows:

$$RMSE = \sqrt{\frac{1}{n}\sum_{i=1}^{n}(y_i - \hat{y}_i)^2} \tag{16}$$

$$MAE = \frac{1}{n}\sum_{i=1}^{n}|y_i - \hat{y}_i| \tag{17}$$

$$R^2 = 1 - \frac{\sum_{i=1}^{n}(y_i - \hat{y}_i)^2}{\sum_{i=1}^{n}(y_i - \overline{y})^2} \tag{18}$$

where $n$ represents the number of observed data in the test data; $y_i$ and $\hat{y}_i$ denote the observed algal parameter values and predicted algal parameter values, respectively. Also, $\overline{y}$ displays the mean observed values.

Moreover, the improvement percentage is introduced to facilitate quantitative comparisons between the decomposition-base hybrid deep learning model and single models. The improvement percentage for the RMSE, MAE, and $R^2$ are calculated as follows:

$$P_{RMSE} = \frac{RMSE_1 - RMSE_2}{RMSE_1} \times 100\% \tag{19}$$

$$P_{MAE} = \frac{MAE_1 - MAE_2}{MAE_1} \times 100\% \tag{20}$$

$$P_{R^2} = \frac{R^2{}_1 - R^2{}_2}{R^2{}_1} \times 100\% \tag{21}$$

where $RMSE_1$ and $MAE_1$ denote the errors of the single models, and $RMSE_2$ and $MAE_2$ represent the errors of the decomposition-based models. $R^2{}_1$ and $R^2{}_2$ denote the errors of the decomposition-based and single models, respectively. A substantial positive value indicates superior accuracy of the decomposition-based models compared to the single models.

**3. Results and Discussion**

Firstly, we present the decomposition results of the WT, EEMD, and SSA models. Secondly, we present and discuss the Chl*a* concentration forecasting enhancement by the hybrid WT-CNN, EEMD-CNN, SSA-CNN, WT-LSTM, EEMD-LSTM, and SSA-LSTM models compared to the independent CNN and LSTM models. Then, we compare the performance among those hybrid deep learning models. Finally, we also assess the strengths and limitations of these decomposition techniques in algal parameter prediction.

### 3.1. The Process of Different Decomposition-Based Algorithms for Chlorophyll Fluorescence Data

Due to the intricate non-stationary nature of algal parameter series, the appropriate decomposition of the original data plays an important role in improving forecasting accuracy. After data preprocessing, the Chl*a* concentration algal parameters were decomposed using the WT, EEMD, and SSA methods. The decomposition results of the algal parameters at the Duanqiao and Caohai center sites are shown in Figure 5. In detail, the wavelet decomposition of the hourly original chlorophyll series effectively generated an approximation low-frequency coefficient (A3) and three detailed high-frequency coefficients (D1–D3) at the Duanqiao and Caohai center sites, as depicted in Figure 5a,b. Compared to the original time series, the sub-series A3 extracted the hourly algal series trend and major peaks, while sub-series levels D1 to D3 captured more subtle and fluctuation components simultaneously.

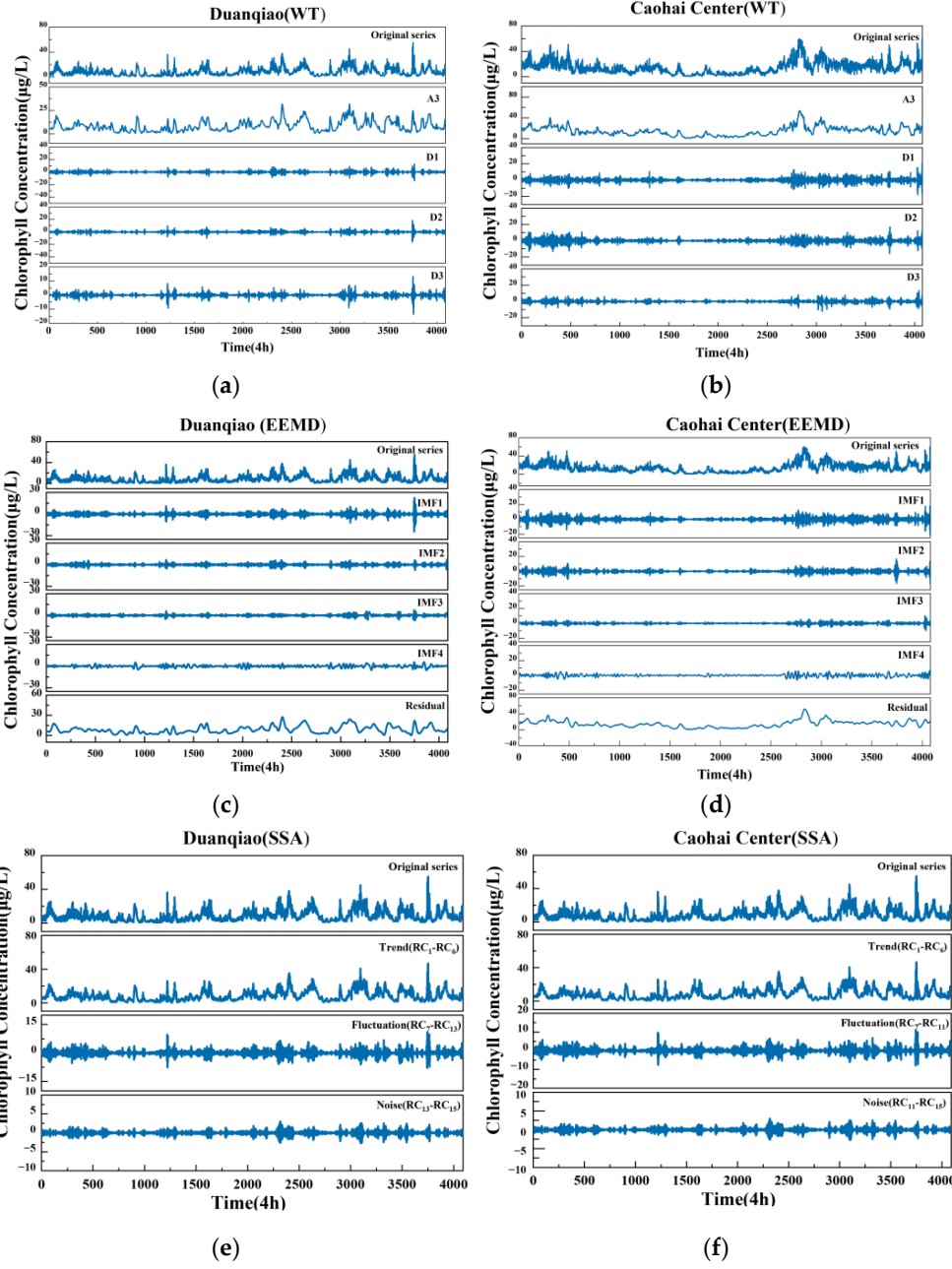

**Figure 5.** (**a**) Decomposition results of the Chl*a* concentrations for (**a**,**b**) WT; (**c**,**d**) EEMD; (**e**,**f**) SSA in Duanqiao and Caohai center sites, respectively.

The EEMD results of the algal parameters are shown in Figure 5c,d. The hourly original chlorophyll series were decomposed into four volatility characteristic IMFs with different frequencies and one residual component. Figure 5c,d show IMFs from high frequency to low frequency, the residue, and the original algal series sequentially from top to bottom. Intuitively, it is clear that the EEMD captures the trend of chlorophyll series and volatility characteristics exactly. Furthermore, the SSA decomposed the algal series into 15 components (Figure 5e,f) with a window length setting to 15. The $RC_1$–$RC_6$ components, with a contribution rate of 97.50%, were reconstructed into the main trend terms. The $RC_7$–$RC_{13}$ sub-series was chosen as the fluctuation component, and $RC_{14}$–$RC_{15}$ was considered as the noise at the Duanqiao site. In comparison, the top six sub-series (i.e., $RC_1$–$RC_6$) with a contribution rate of 97.90% were chosen as the main trend components of algal parameters, and the remaining $RC_7$–$RC_{11}$ and $RC_{11}$–$RC_{15}$ components were considered as the volatility characteristics and noise into the subsequent model at the Caohai center site. The series of algal parameters exhibited more prominent trends after reconstruction, suggesting that SSA can effectively extract the trend, volatility, and noise components, capturing the primary features of the series.

### 3.2. Evaluating the Predictive Performance of Deep Learning Based on Multi-Decomposition Process

In this section, the CNN and LSTM predict each sub-series obtained through the multi-decomposition methods. We procure the final algal series' predicted values by summing up the forecasting results of all the sub-components. To adequately assess the effectiveness of the hybrid deep learning models, we employ hybrid CNN and LSTM models based on multi-decomposition, including WT-CNN, EEMD-CNN, SSA-CNN, WT-LSTM, EEMD-LSTM, and SSA-LSTM. The detailed performance curves of the prediction and observed values in the train and test datasets are shown in Figure 6. From Figure 6, the fitting curve obtained by the SSA-based hybrid models (orange lines) closely approximates the observed values (blue lines), especially pronounced at peak locations. Therefore, it implies that the performance of SSA-based hybrid models are better than the performance of WT-based and EEMD-based hybrid models in algal sequence prediction. While individual CNN and LSTM models demonstrate the ability to forecast the Chl*a* trends, significant errors exist between observed and predicted values. These models demonstrate inadequate prediction accuracy in capturing details and sharp peaks. The decomposition-based hybrid models can more accurately predict the detailed components and significantly enhance the model performance. All of this evidence convincingly indicates that the decomposition-based hybrid methods can reliably predict algal dynamics at different sampling sites.

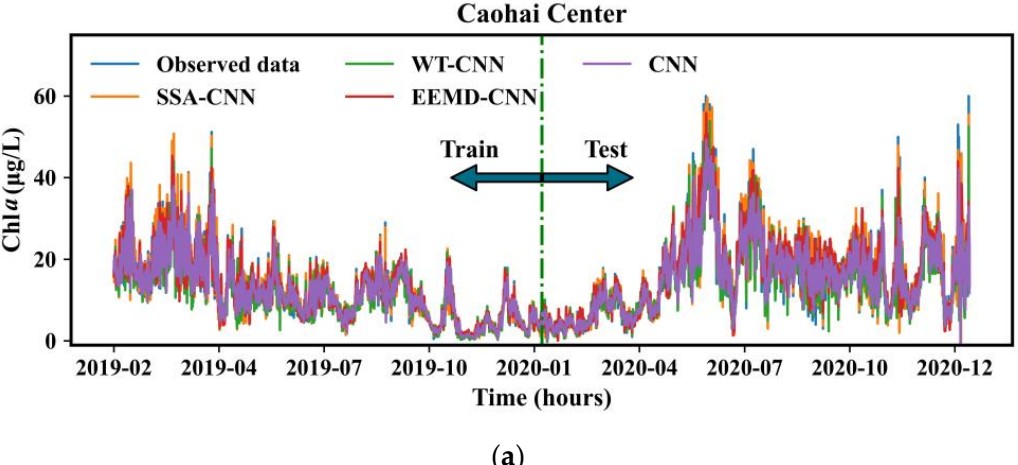

(a)

**Figure 6.** *Cont.*

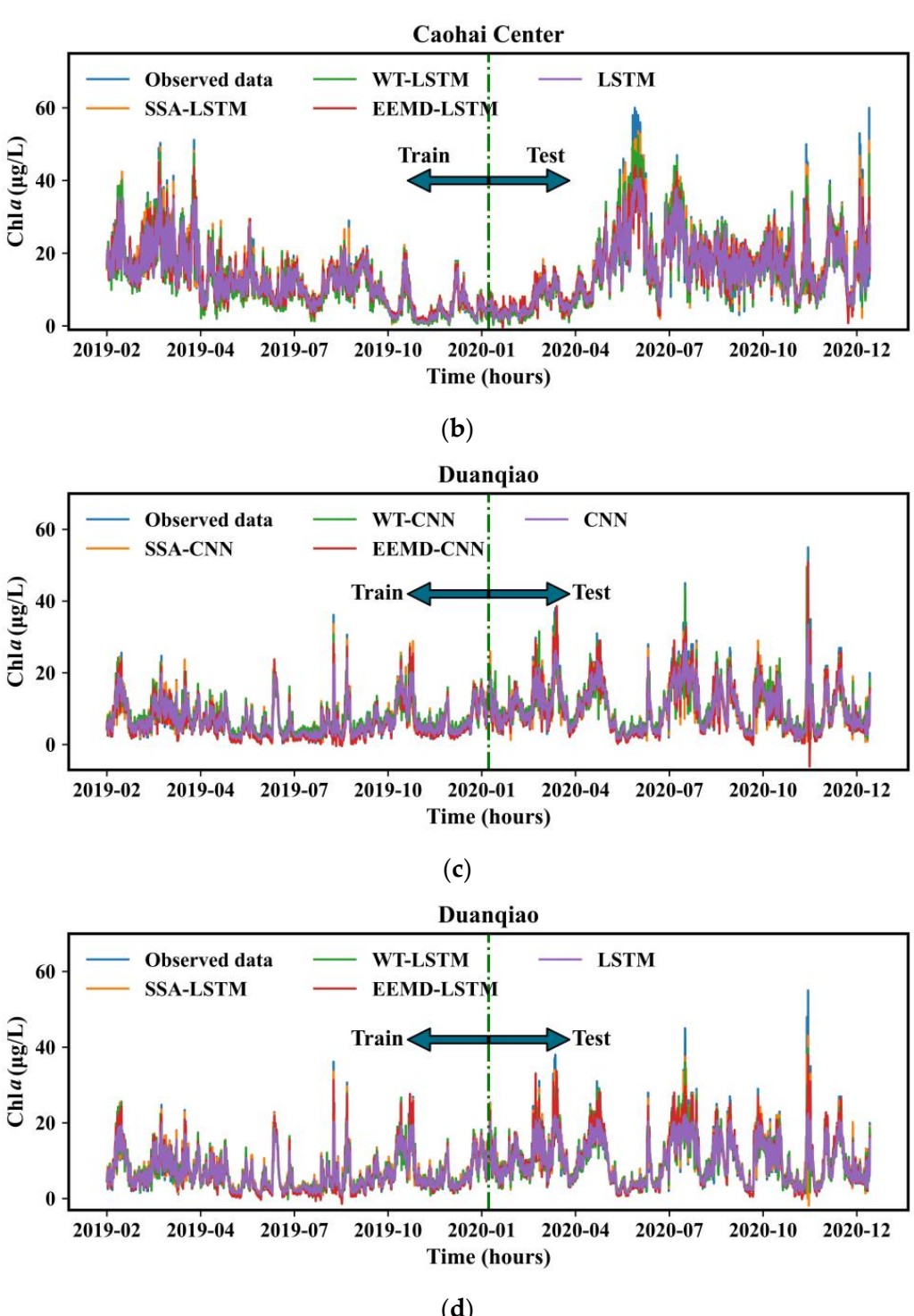

**Figure 6.** Predicted and observed time series of Chl*a* concentrations by the decomposition-based hybrid deep learning models and independent CNN and LSTM approaches in the Lake Dianchi at (**a**,**b**) Caohai Center and (**c**,**d**) Duanqiao.

The hybrid decomposition-based deep learning framework and commonly employed time series individual forecasting methods (CNN and LSTM) are also cross-compared based on the Caohai center and Duanqiao sites of Lake Dianchi (Figure 7). For the CNN model, the best predictions were achieved by the SSA-based hybrid model, resulting in highly satisfactory $R^2$ values ($R^2$ = 0.9652 and 0.9518), which represented a 32.16% increase for the Caohai center site and a 26.67% increase for the Duanqiao site (Figure 7a,b). When

compared to the CNN, the prediction accuracy of WT-CNN and EEMD-CNN, as measured based on the evaluation indicators of $R^2$, showed improvements of 28.02%, 24.99%, 19.18%, and 14.8% for the two sites, respectively. Furthermore, as for the LSTM approach, which is consistent with the CNN method, SSA-LSTM outperforms the single LSTM in terms of its $R^2$ statistic (0. 978 vs. 0.738 in Caohai center, and 0.975 vs. 0. 723 in Duanqiao, respectively). Additionally, although the WT-LSTM models ($R^2$ = 0.955, 0.944) perform better than EEMD-LSTM ($R^2$ = 0.857, 0.881) at the two sites, they still exhibit a slightly lower performance compared to the SSA-LSTM approach. As for the CNN and LSTM approaches, all models demonstrated the lowest performance among the four cases examined (Figure 7), showing the limitation of single deep learning models in handling complicated and non-stationary forecasting tasks of algal dynamics.

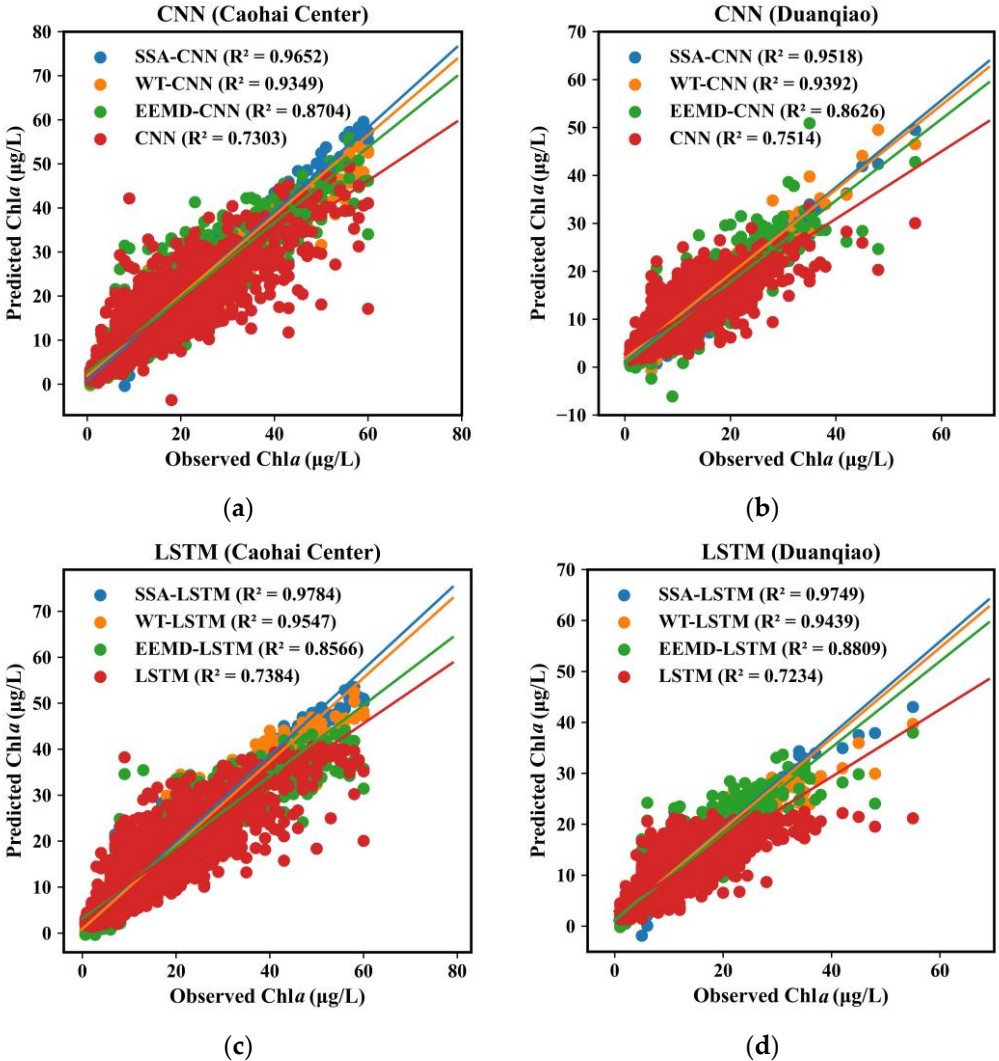

**Figure 7.** Scatter diagrams of the Chl*a* with one-step prediction lead times at the Caohai center and Duanqiao stations in Lake Dianchi. Red dots represent the single models, green dots represent the hybrid EEMD-based models, orange dots represent the hybrid WT-based models, and blue dots represent the hybrid SSA-based models. (**a**) Comparison of single model and hybrid CNN-based models at Caohai center station. (**b**) Comparison of single model and hybrid CNN-based models at Duanqiao station. (**c**) Comparison of single model and hybrid LSTM-based models at Caohai center station. (**d**) Comparison of single model and hybrid CNN-based models at Duanqiao station.

Cross-comparisons between hybrid-based CNN or LSTM and single models show that the most significant improvement in forecasting HAB dynamics is achieved through decomposition-based hybrid approaches. These outcomes show that the hybrid deep learning algorithms based on multi-decomposition can decompose non-stationary time series into sub-sequences and better detect the main trend, volatility, and noise components of algal dynamics. In addition, sub-series are fed to the CNN and LSTM models one by one to improve HAB forecasting further. The hybrid models based on decomposition exhibit the ability to uncover the peak and extreme values of algal parameters, indicating the robust resilience and effective signal-smoothing capability of the preprocessing approach. This is in line with the previous study by Liu et al. [26], who demonstrated that the decomposition-based hybrid WT-LSTM model could enhance the fitting of abrupt or extreme data points and improve HAB prediction performance. Likewise, Luo et al. [30] developed a hybrid EEMD-LSTM model for predicting water quality, and the results demonstrated that the proposed model outperformed the individual LSTM model in various evaluation indicators. Cui et al. [42] also discovered that integrating SSA with a lightweight gradient-boosting machine in a hybrid model led to high-accuracy, real-time predictions of urban runoff. Interestingly, the advantages of hybrid models integrating LSTM over CNN-based models become apparent, indicating the benefits of automatically capturing long-temporal information through LSTM recurrent chains.

### 3.3. Comparing the Effectiveness of Different Decomposition Approaches in Forecasting HABs

To compare the prediction performance of different decomposition approaches, we combine common decomposition approaches—namely WT, EEMD, and SSA—with the CNN and LSTM forecasting techniques to forecast the algal series at two stations. The results are presented in Tables 3 and 4. Firstly, when combined with the SSA decomposition, the hybrid-based LSTM and CNN prediction methods reach the lowest RMSE (1.518 μg/L, 1.927 μg/L, 1.092 μg/L and 1.513 μg/L) and MAE values (0.855 μg/L, 1.361 μg/L, 0.702 μg/L, 1.025 μg/L) and the biggest $R^2$ (0.978, 0.965, 0.975, 0.952) at Caohai center and Duanqiao station, respectively. This verifies the superiority of the SSA decomposition method over other WT and EEMD methods. Secondly, the decomposition methods demonstrate enhanced effectiveness when combined with a more precise forecasting technique. In particular, LSTM can achieve significantly improved accuracy percentages through decomposition compared to the CNN. Specifically, WT decomposition increases the LSTM improvement percentages of the RMSE from 5.282 to 2.198 by 58.4%, the MAE from 3.525 to 1.405 by 60.1%, and the $R^2$ from 0.738 to 0.955 by 29.3%; the CNN improvement percentages of RMSE increase from 5.364 to 2.635 by 50.9%, MAE from 3.538 to 1.742 by 50.8%, and the $R^2$ from 0.730 to 0.935 by 28%. Similarly, the SSA decomposition method offers a greater performance enhancement for LSTM than the CNN model, while EEMD decomposition yields similar results for both models. Thirdly, employing the same WT and SSA decomposition techniques, LSTM typically outperforms the CNN regarding prediction accuracy, while for EEMD decomposition, LSTM is not always superior to the CNN.

**Table 3.** The improvement percentages in the accuracy of the decomposition-based CNN and LSTM methods compared with single CNN and LSTM approaches for Chl*a* (chlorophyll-*a*) prediction in Caohai center station (µg/L). (+Δ represents the improvement percentages in RMSE and MAE for decomposition-based models compared to single models.)

| Models | Training Set | | | | | | Test Set | | | | | |
|---|---|---|---|---|---|---|---|---|---|---|---|---|
| | RMSE (µg/L) | | MAE (µg/L) | | $R^2$ | | RMSE (µg/L) | | MAE (µg/L) | | $R^2$ | |
| LSTM | 3.435 | | 2.324 | | 0.813 | | 5.282 | | 3.525 | | 0.738 | |
| EEMD-LSTM | 2.143 | (+Δ37.6%) | 1.475 | (+Δ36.6%) | 0.927 | (+Δ14.0%) | 3.912 | (+Δ26.0%) | 2.383 | (+Δ32.4%) | 0.857 | (+Δ16.0%) |
| WT-LSTM | 1.255 | (+Δ63.5%) | 0.888 | (+Δ61.8%) | 0.975 | (+Δ19.9%) | 2.198 | (+Δ58.4%) | 1.405 | (+Δ60.1%) | 0.955 | (+Δ29.3%) |
| SSA-LSTM | 0.740 | (+Δ78.5%) | 0.491 | (+Δ78.9%) | 0.991 | (+Δ21.9%) | 1.518 | (+Δ71.3%) | 0.855 | (+Δ75.7%) | 0.978 | (+Δ32.5%) |
| CNN | 3.405 | | 2.323 | | 0.816 | | 5.364 | | 3.538 | | 0.730 | |
| EEMD-CNN | 2.122 | (+Δ37.7%) | 1.477 | (+Δ36.4%) | 0.929 | (+Δ13.8%) | 3.719 | (+Δ30.7%) | 2.461 | (+Δ30.4%) | 0.870 | (+Δ19.2%) |
| WT-CNN | 1.499 | (+Δ56.0%) | 1.035 | (+Δ55.4%) | 0.964 | (+Δ18.1%) | 2.635 | (+Δ50.9%) | 1.742 | (+Δ50.8%) | 0.935 | (+Δ28.0%) |
| SSA-CNN | 1.266 | (+Δ62.8%) | 0.900 | (+Δ61.3%) | 0.975 | (+Δ19.4%) | 1.927 | (+Δ64.1%) | 1.361 | (+Δ61.5%) | 0.965 | (+Δ32.2%) |

**Table 4.** The improvement percentages in the accuracy of the decomposition-based CNN and LSTM methods compared with single CNN and LSTM approaches for Chl*a* (chlorophyll-*a*) prediction in Duanqiao station (µg/L). (+Δ represents the improvement percentages in RMSE and MAE for decomposition-based models compared to single models.)

| Models | Training Set | | | | | | Test Set | | | | | |
|---|---|---|---|---|---|---|---|---|---|---|---|---|
| | RMSE (µg/L) | | MAE (µg/L) | | $R^2$ | | RMSE (µg/L) | | MAE (µg/L) | | $R^2$ | |
| LSTM | 2.381 | | 1.588 | | 0.739 | | 3.624 | | 2.295 | | 0.723 | |
| EEMD-LSTM | 1.281 | (+Δ46.2%) | 0.913 | (+Δ42.5%) | 0.924 | (+Δ25.1%) | 2.378 | (+Δ34.4%) | 1.452 | (+Δ36.7%) | 0.881 | (+Δ21.8%) |
| WT-LSTM | 0.855 | (+Δ64.1%) | 0.595 | (+Δ62.6%) | 0.966 | (+Δ30.8%) | 1.633 | (+Δ54.9%) | 0.942 | (+Δ58.9%) | 0.944 | (+Δ30.5%) |
| SSA-LSTM | 0.650 | (+Δ72.7%) | 0.543 | (+Δ65.8%) | 0.981 | (+Δ32.7%) | 1.092 | (+Δ69.9%) | 0.702 | (+Δ69.4%) | 0.975 | (+Δ34.8%) |
| CNN | 2.444 | | 1.680 | | 0.725 | | 3.436 | | 2.251 | | 0.751 | |
| EEMD-CNN | 1.281 | (+Δ47.6%) | 0.931 | (+Δ44.6%) | 0.924 | (+Δ27.6%) | 2.554 | (+Δ25.7%) | 1.614 | (+Δ28.3%) | 0.863 | (+Δ14.8%) |
| WT-CNN | 1.309 | (+Δ46.4%) | 1.116 | (+Δ33.6%) | 0.921 | (+Δ27.1%) | 1.699 | (+Δ50.6%) | 1.284 | (+Δ43.0%) | 0.939 | (+Δ25.0%) |
| SSA-CNN | 0.858 | (+Δ64.9%) | 0.691 | (+Δ58.8%) | 0.966 | (+Δ33.3%) | 1.513 | (+Δ56.0%) | 1.025 | (+Δ54.5%) | 0.952 | (+Δ26.7%) |

Table 5 provides a concise comparison of the WT, EEMD, and SSA decomposition methods employed in this study. It is essential to note that each approach possesses its distinct strengths and limitations. Specifically, WT is commonly employed as a preprocessing method for predicting water levels [43], algal blooms [26], precipitation [44], rainfall runoff [45], and river flow [46]. WT is well suited for handling signals with a constant frequency and near periodicity, but requires presetting the basis function and their order, significantly impacting the decomposition results. The noise-assisted ensemble empirical mode decomposition (EEMD) method is utilized to address the issue of mode mixing and is capable of processing complex and non-stationary time series. EEMD has gained extensive adoption in enhancing the forecasting performance of precipitation [47], daily runoff [48], water levels [49], rainfall [50], streamflow [51], river flow [52], and water quality [30]. Singular spectral analysis (SSA) has found widespread applications in the preprocessing of hydrological data, including streamflow [31], rainfall [53], runoff [54], and rainfall runoff [42] prediction. Previous research has demonstrated that SSA can significantly enhance the prediction effectiveness of independent deep learning models, highlighting its potential to enhance forecasting accuracy. Diverse decomposition methods exhibit distinct applicability conditions. Consequently, selecting the appropriate decomposition methods based on different applicability ranges can provide a research example for the prediction of algal blooms. Despite the extensive research on decomposition-based

hybrid deep learning models, some obstacles still demand additional consideration. Firstly, the decomposition process includes future unknown data, leading to information leakage. Secondly, the predictability of individual sub-series can vary, leading to differences in the weighting of prediction residuals among them. Thus, selecting suitable decomposition and prediction methods based on varying application scenarios is of significance in algal bloom forecasting, thereby contributing to the early warning and management of algal blooms.

**Table 5.** The concise comparison of decomposition models for different variable forecasting.

| Methods | Advantages | Disadvantages | Variables |
|---|---|---|---|
| Wavelet Transform (WT) | Strict mathematical theory. Appropriate for steady-frequency and nearly periodic signals. | Requires the prior specification of wavelet basis and parameters. Separates the modes. Not suitable for highly non-stationary signals. | Water level [43] Algal bloom [26] Precipitation [44] Rainfall runoff [45] River flow [46] |
| Ensemble Empirical Mode Decomposition (EEMD) | Fully data-driven. Addresses mode mixing issue. Suitable for both non-linear and non-stable signals. Fully adaptive by originally introducing the intrinsic mode functions (IMFs). | Lacks rigorous mathematical theory. Additional noise is present in the reconstructed signal. Needs many computational resources. | Precipitation [47] Daily Runoff [48] Water Level [49] Rainfall [50] Streamflow [51] River flow [52] Water Quality [30] |
| Singular Spectrum Analysis (SSA) | Strict mathematical theory. Decompose a time series into distinct components. | Parameters must be fine-tuned to isolate each component. | Streamflow [31] Rainfall [53] Runoff [54] Rainfall runoff [42] |

## 4. Conclusions

To attain high-performance chlorophyll fluorescence forecasting results, hybrid chlorophyll prediction models coupling multi-decomposition methods were developed in this paper. Firstly, wavelet transform (WT), the empirical mode decomposition model (EEMD), and singular spectral analysis (SSA) are employed on time-series sequences of hourly chlorophyll data of Lake Dianchi to decompose original sequences into multiple sub-series, including high-frequency and low-frequency components simultaneously. Moreover, for individual forecasting, all sub-sequences are fed to deep learning models, namely a convolutional neural network (CNN) and long short-term memory (LSTM). Ultimately, the predicted chlorophyll values are obtained by summing up the results of the forecasting sub-series. The results illustrate that decomposition methods can precisely capture the trend, volatility, and noise of chlorophyll fluorescence, overcoming the shortcomings of the independent deep learning model in processing non-linear chlorophyll sequences. Furthermore, the chlorophyll prediction results from two test stations indicate that hybrid deep learning models based on decomposition achieve higher prediction accuracy than single approaches, with the improvement percentages of the RMSE increasing, ranging from 25.7% to 71.3%, with MAE improvements ranging from 28.3% to 75.7% and the $R^2$ values increasing by 14.8% to 34.8%. In addition, the results of the comparative experiment involving various decomposition techniques substantiate SSA decomposition's superiority over the commonly employed WT and EEMD methods for chlorophyll forecasting. In conclusion, the decomposition–prediction framework has the potential for applications in short-term prediction of HABs by implementing high-frequency monitoring. This allows for the anticipation of future trends in algal blooms, serving as a foundation for devising measures to manage the water environment.

**Author Contributions:** Conceptualization, L.W., M.S. and K.S.; Methodology, L.W., M.X., Z.G., X.W. and K.S.; Software, Z.G.; Validation, L.W., M.X., B.Y., X.W. and K.S.; Investigation, L.W., M.X., M.P., F.H., B.Y., X.W. and K.S.; Resources, M.P., F.H. and K.S.; Data curation, L.W., M.X., M.P., F.H., B.Y.,

Z.G. and K.S.; Writing—original draft, L.W., M.X. and K.S.; Writing—review & editing, L.W., M.X. and K.S.; Supervision, L.W., M.P., F.H., B.Y., Z.G. and M.S.; Project administration, M.P., F.H., Z.G., M.S. and K.S.; Funding acquisition, M.S., K.S., L.W. and B.Y. All authors have read and agreed to the published version of the manuscript.

**Funding:** This work was supported by the Yunnan Science and Technology Commission (2022 02AH210006-4), National Natural Science Foundation of China (No. 52379081; 62072429), the Chongqing Science and Technology Commission (CSTB2022TIAD-KPX0199; cstc2021jcyj-msxmX1199), the Chongqing Education Commission (HZ2021008; KJQN202101622), the West Light Foundation of The Chinese Academy of Sciences (E1296001).

**Data Availability Statement:** The data presented in this study are available on request from the corresponding authors.

**Conflicts of Interest:** The authors declare no conflict of interest.

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
