# Peer review of "Improved Deep Learning Predictions for Chlorophyll Fluorescence Based on Decomposition Algorithms: The Importance of Data Preprocessing"

_water, doi:10.3390/w15234104_

Round 1
Reviewer 1 Report
Comments and Suggestions for Authors
This paper compared the performance of multiple hybrid approaches coupling decomposition and deep-learning techniques for the prediction of harmful algal blooms presence in lakes of China, also utilizing the use of frequently Chl-a field acquired timeseries of data for the training and validation of the models/algorithms. The use of English is excellent, although authors should apply a last check for minor typos and syntax mistakes (a number of them is listed below for the authors convenience). Literature analysis, design of methods and presentation/discussion of the results are of very high standards and quality, supported by the illustration of relevant plots, graphs, and tables. Although there is no novelty regarding the formulation and design of the adapted techniques, the combination of methods proposed is novel and illustrates significant and robust results, while all technical aspects considering the methods and mathematical models adapted are clearly described. Comparison with relevant studies and their results is adequately analysed in the Results and discussion section, whereas limitations of the study are also identified. Conclusions expressed highlight the findings of the study and demonstrate significant scientific importance. In overall, it is suggested to accept this high-quality paper for publication, after minor additions listed below:
1. Please add Line Numbering, it makes it difficult to refer to specifc parts of the manuscript.
2. Introduction Section - "Following that, integrated SSA and genetic models -based have been developed...." change to "genetic-based models" or rephrase to improve syntax.
3. Section 2.3 Model implementation - ".... 50% of the decomposed sequence (Feb. 2019–Jan. 2020) are as the input tobuild the network." a verb seems to be missing between "are as".
4. Section 3.2. Evaluation the predictive performance of deep learning based on multi-decomposition process - Change title to "Evaluation of the..." or to "Evaluating the..."
5. Section 3.2 - "....LSTM models demonstrat the ability...." correct typo to "demonstrate".
6. Section 3.2 - "All of this evidence convincingly indicat that the...." correct typo to "indicate".
7. Section 3.2 - "....at the two sites, it still exhibit" change "it" to "they", since it refers to "models".
8. Section 3.2 - "....show that the most significant forecasting improvement of HAB dynamics by hybrid approaches decomposition-based. " Sentence needs to be rephrased to improve syntax.
9. Section 3.2 - "....and effective signal-smoothing capability of the preprocessing approach...." try to be consistent throughout the manuscript for terms use: in this case, choose between "pre-processing" or "preprocessing" and apply it to the whole manuscript. Same comment applies to other terms too, e.g., R2 and R2, please apply a detailed check for such incosistencies in the manuscript and correct them accordingly.
10. Section 3.3. Comparisons the effectiveness of different decomposition approaches in forecasting HABs - Change title to "Comparison of the..." or to "Comparing the...".
11. Section 3.3 - "Specifically, WT decomposition increases the RMSE of LSTM from 5.282 to 2.198 by58.4%, MAE from 3.525 to 1.405 by 60.1% and the R2 from 0.738 to 0.955....." if I understand correctly what is described, you should change"increases" to "decreases" when referring to RMSE and MAE, and keep "increases" in case of R2, at least according to my understanding of the results presented in the table.
12. Table 4 Caption - "he comparison on various...." change "he" to "The".
Comments on the Quality of English LanguageIncluded above
Reviewer 2 Report
Comments and Suggestions for Authors
Review of the article "Improved deep-learning predictions for Chlorophyll fluorescence based on decomposition algorithms: importance of data preprocessing"
In my opinion, the article is interesting and valuable.
My points to improve the article:
1. In the Abstract the informatic features (i.e. the IT values) have been emphasized. Moreover, there are presented information that cannot be evaluated (or is difficult to evaluate) by readers before reading the article, i.e.
"The results indicate that decomposition-based models can enhance the prediction accuracy of single deep learning, in terms of RMSE increasing ranging from 25.7% to 71.3%, MAE ranging from 28.3% to 75.7%, and R2 values increasing ranging from 14.8% to 34.8%."
In the Abstract should be presented and emphasized the aim of the work that has to be in accordance with the scope of the Water journal (https://www.mdpi.com/journal/water/about).
2. In the Introduction is:
"However, HABs exhibit uneven distributions with significant inter-annual and seasonal variability, and numerous drivers interactively influence cyanobacterial abundance and the formation of HABs across differing spatial and temporal scales[2]. Therefore, additional modelling efforts are necessary to overcome the challenges associated with HABs forecasting."
The reason for the need for additional modeling that increase effectiveness of HABs forecasting is that existing HABs forecasting models established by the other authors do not give satisfactory results.
For this reason, existing solutions (i.e. presented by the other authors) of modeling that enable HABs forecasting should be presented in the Introduction. This information should indicate that existing solutions do not meet expectations and therefore there is a need to continue research in this area in order to increase the effectiveness of modeling.
3. In the Introduction is:
"Specifically, the goals of this paper are as follows:
1) to validate the performance improvement of WT, EEMD and SSA decomposition approaches;
2) to evaluate the prediction effectiveness of decomposition-based hybrid deep leaning models;
3) to further compare the prediction performance of different decomposition approaches for HABs forecasting.
This study demonstrates that the decomposition based hybrid deep-learning methods could function as a robust and trustworthy tool for forecasting HABs in water management."
The goal of the work should be more clearly related to the scope of the Water journal (https://www.mdpi.com/journal/water/about). From the Introduction it can be concluded that this is an article related to computer science (informatics) and submitted to informatic journal.
4. There should be added text after the "2.2 Decomposition-based deep-learning models development" section.
5. The subsections from 2.2.1 to 2.2.6 should be moved to Appendix as supporting, supplementary information because it is not directly related to the scope of the Water journal.
6. In Table 2 the values of momentum, the input size and output size should also be presented.
7. The Conclusions section contains no conclusions directly related to the topic of the Water journal (https://www.mdpi.com/journal/water/about).
8. Linguistic mistakes (including typographical) require correction.
Comments on the Quality of English Language
Linguistic mistakes (including typographical) require correction.
Round 2
Reviewer 2 Report
Comments and Suggestions for Authors
The Authors have correctly addressed all my concerns and comments. Now, the article is better (including better fitting to the scope of the journal) and in my opinion it can be published in Water.